# A Novel Zwitterionic Hydrogel Incorporated with Graphene Oxide for Bone Tissue Engineering: Synthesis, Characterization, and Promotion of Osteogenic Differentiation of Bone Mesenchymal Stem Cells

**DOI:** 10.3390/ijms24032691

**Published:** 2023-01-31

**Authors:** Qidong Wang, Meng Li, Tianming Cui, Rui Wu, Fangfang Guo, Mei Fu, Yuqian Zhu, Chensong Yang, Bingdi Chen, Guixin Sun

**Affiliations:** 1Department of Traumatic Surgery, Shanghai East Hospital, School of Medicine, Tongji University, Shanghai 200120, China; 2Shanghai Research Institute for Intelligent Autonomous Systems, Tongji University, Shanghai 200092, China; 3School of Materials Science and Engineering, Tongji University, Shanghai 201804, China; 4The Institute for Biomedical Engineering & Nano Science, School of Medicine, Tongji University, Shanghai 200092, China

**Keywords:** zwitterionic, graphene oxide, osteogenic differentiation, stem cell, tissue engineering

## Abstract

Zwitterionic materials are widely applied in the biomedical field due to their excellent antimicrobial, non-cytotoxicity, and antifouling properties but have never been applied in bone tissue engineering. In this study, we synthesized a novel zwitterionic hydrogel incorporated with graphene oxide (GO) using maleic anhydride (MA) as a cross-linking agent by grafted L-cysteine (L-Cys) as the zwitterionic material on maleilated chitosan via click chemistry. The composition and each reaction procedure of the novel zwitterionic hydrogel were characterized via X-ray diffraction (XRD) and Fourier transformed infrared spectroscopy (FT-IR), while the morphology was imaged by scanning electron microscope (SEM). In vitro cell studies, CCK-8 and live/dead assay, alkaline phosphatase activity, W-B, and qRT-CR tests showed zwitterionic hydrogel incorporated with GO remarkably enhanced the osteogenic differentiation of bone mesenchymal stem cells (BMSCs); it is dose-dependent, and 2 mg/mL GO is the optimum concentration. In vivo tests also indicated the same results. Hence, these results suggested the novel zwitterionic hydrogel exhibited porous characteristics similar to natural bone tissue. In conclusion, the zwitterionic scaffold has highly biocompatible and mechanical properties. When GO was incorporated in this zwitterionic scaffold, the zwitterionic scaffold slows down the release rate and reduces the cytotoxicity of GO. Zwitterions and GO synergistically promote the proliferation and osteogenic differentiation of rBMSCs in vivo and in vitro. The optimal concentration is 2 mg/mL GO.

## 1. Introduction

Large segmental bone defects and bone nonunion caused by trauma, tumor, and infection remain a major tricky challenge for orthopedic surgeons [1,2]. Autogenic and allogenic bone graft transplantation are considered effective treatments due to faster, efficient healing [3,4]. However various problems, such as source limitation, immune refection, and donor site morbidity, hinder its wide application. Therefore, it is necessary to fabricate a synthetic bone graft substitute to satisfy the great demand in the clinic. Zwitterionic materials have attracted extensive attention in the biomedical field due to their excellent antimicrobial, non-cytotoxicity [5], and antifouling properties [6,7]. They have equal amounts of anionic and cationic groups on one molecule and present as electric neutral, which endows them with a strong ability to bind water molecules via electrostatic interactions and enhances resistance to non-specific adhesion of proteins, molecules, and cells [8,9,10,11]. Amino acids, as naturally zwitterionic compounds, own the zwitterionic group of carboxyl (–COOH) amine (–NH2) [12,13]. As one of the amino acids, L-cysteine has good hydrophilicity and promotes cell proliferation. Overall, it is possessed by the thiol group, which can act as a reactive group to click on carbon–carbon double bonds, such as maleic anhydride (MA), via click chemistry.

Chitosan (CS), a natural polysaccharide derived from the deacetylation of chitin, has been widely applicated in bone tissue engineering for its unique benefits of easy availability, good antibacterial properties [14], versatile functionality, biocompatibility, and biodegradability [15]. However, the potential of natural materials is limited by their relatively low mechanical strength and rapid degradation. Zhou et al. have successfully fabricated a new hydrogel via crosslinked MA on CS [16]. In this study, we want to synthesize a novel hydrogel using MA as a crosslinking agent; L-cysteine was grafted on maleilated chitosan via click chemistry to form a stable three-dimensional hydrogel network.

Graphene oxide (GO) is an oxygen-containing derivative of graphene with a large surface area consisting of a two-dimensional carbon honeycomb structure. It is rich in hydroxyl functional groups and carboxylic groups on the surface. This hydrophilic character makes GO extremely dissolved in water and organic solvents and easily connected with biomaterials [17], promoting cell adhesion and proliferation [18]. GO can induce osteogenic differentiation of MSCs without adding osteogenic induction supplements [19]. The cytotoxicity of GO is dose dependent, and the interaction between GO and rBMSCs is dependent on size, dose, and topography [20]. Incorporation of GO with various hydrogels can enhance their physicochemical and biological features [21,22,23,24] and have been widely used in bone tissue engineering [25,26]. Some studies reported GO incorporated in CS-based hydrogel can improve the mechanical properties and promote BMSC osteogenic differentiation [27,28]. Moreover, GO incorporation can improve cell adhesion, spreading, proliferation, and the osteogenic differentiation of stem cells [29,30].

In human bones, the most inorganic matter is hydroxyapatite (HA). In contrast to sintered HA, β-tricalcium phosphate (β-TCP) has a similar structure, but has excellent biodegradability in vivo. It can degrade into calcium and phosphate ions in the body, which play important roles in stimulating new bone formation [31,32,33]. Some studies have reported the application of composites consisting of β-TCP with natural polymers, such as chitosan/gelatin [34,35] and dextran [36]. Wang et al. proved porous (β-TCP) scaffolds promote cell migration, proliferation, and vascularization [37]. Liu et al. reported β-TCP and GO coordination facilitates BMSC osteogenesis [38].

In this study, we synthesized a novel zwitterionic hydrogel using MA as the crosslinking agent and grafted the zwitterionic material on maleilated chitosan via click chemistry. The zwitterionic scaffold has strong mechanical properties and a low-degradation rate, which could slow down the release rate and reduce the cytotoxicity of GO. We incorporated different concentrations of GO (0, 1, 2, and 4 mg/mL) into the zwitterionic hydrogel and evaluated the effect of GO on mechanical properties of zwitterionic hydrogel and rBMSCs proliferation and osteogenesis.

## 2. Results and Discussion

### 2.1. Morphology Characterization of Zwitterionic Hydrogel Scaffolds

Figure 1A showed the photo image of zwitterion hydrogel with various concentrations of GO. Without GO, the hydrogel was milky white, and the color of the hydrogel scaffolds changed gradually, darkening from milky white to brown to black with the increase in GO concentration. SEM images (Figure 1C) showed the morphology of zwitterion hydrogel presented a honeycomb-like structure and exhibited many particles on the wall. The pore size of the Z-CS/β-TCP hydrogel was uniform, and the pore wall was slightly folded. With the concentration of GO increasing, the porous surface increased in roughness and interconnectivity. More GO sheets were observed, indicating GO sheets loaded with β-TCP particles were evenly dispersed on the wall via van der Waals force. The pore size and interconnectivity are important for the diffusion of nutrients and elimination of metabolic waste [39,40], and the roughness of the wall is beneficial for cell attachment and proliferation [41,42]. The results indicated the addition of GO improves surface roughness and pore interconnectivity, which are vital to promote cell adhesion, spreading, proliferation, and mineralization [30,43].

### 2.2. Porosity, Swelling, Degradation Rate Studies

The porosity of hydrogel is important for the seeding, penetration, distribution, and growth of cells [44,45]. As shown in Figure 1B, the Z-CS/β-TCP/GO-1 hydrogel showed a porosity ratio of 57.33% with no significant difference compared to the Z-CS/β-TCP hydrogel; with increasing GO concentration, the porosity rate of the scaffold increased from 65.17 ± 2.309 in Z-CS/β-TCP/GO-2 hydrogel to 72.33 + 1.732 in Z-CS/β-TCP/GO-4 hydrogel. The Z-CS/β-TCP/GO-4 hydrogel has the highest porosities. The porosities of the scaffolds increased with increased GO concentration [46].

The water absorption of the scaffolds plays an important role in nutrient transport and the discharge of metabolic waste. As shown in Figure 1D, the swelling ratio showed an opposite changing tendency to porosity rate. In the first hour, all samples exhibited a quick liquid absorption capacity. The Z-CS/β-TCP hydrogel scaffold has the highest swelling ratio of 403%, and compared to other scaffolds, the swelling ratios were about 378%, 368%, 340%, and 346%, respectively. However, after one hour, the swelling ratios of all hydrogel scaffolds appeared to have a slow upward trend. At 12 h, the swelling gradually tended to be constant. The swelling ratio of the Z-CS/β-TCP/GO-4 hydrogel scaffold changed from 327% at 0.5 h to 375% at 24 h, which was significantly lower than the Z-CS/β-TCP hydrogel scaffold (378% at 0.5 h to 453% at 24 h). The zwitterionic hydrogel expresses excellent water absorption ability, and L-cysteine increases the hydrophilicity of maleilated chitosan. However, it decreased with an increasing in the percentage of GO, which may be due to the interaction between the zwitterionic hydrogel and the graphene sheets weakening the hydrophilicity of zwitterionic hydrogel [47].

The degradation rate of the zwitterionic hydrogel scaffolds was estimated by detecting the weight loss of scaffolds in fluids at different times. According to Figure 1E, on day three, the weight loss of the Z-CS/β-TCP hydrogel scaffold was higher than Z-CS/β-TCP/GO-4, whereas there were no differences between other groups. The degradation degrees of the hydrogel scaffolds gradually increased with time. After 28 days, a significant change was observed in each group, and the weight loss percentages were 33.58 + 0.739, 26.30 + 1.067, 21,94 + 0.545, 15.40 + 0.661, respectively; the Z-CS/β-TCP/GO-4 hydrogel scaffold was significantly lower than others. This finding suggested a low-degradation rate was associated with a high concentration of GO. The incorporation of GO in zwitterionic hydrogel remarkably slowed down weight loss [48].

### 2.3. Mechanical Properties

The better mechanical properties of biomaterial essentially influence on the growth and differentiation of rBMSCs. As shown in Figure 2A, the stress–strain curves were enhanced with the addition of GO 1 mg/mL. In Figure 2B, the Young’s modulus of the Z-CS/β-TCP/GO-1 hydrogel scaffold was 1.71 MPa, about 36% higher than the Z-CS/β-TCP hydrogel scaffold without GO (1.25 MPa). However, with the concentration of GO beyond 2, 4 mg/mL would reduce the mechanical properties of the scaffolds and deteriorate their mechanical performance. The GO sheets in the zwitterionic hydrogel enhanced the mechanical properties via the H-bonding interaction between GO and CS. Nevertheless, the content of GO beyond 2 mg/mL in zwitterionic hydrogel would diminish the mechanical properties of the hydrogel, owning to the restacking and aggregation of the GO sheets [49,50].

### 2.4. Structure Characterization of GO Scaffolds

The FT-IR spectrum revealed the reaction of the zwitterionic hydrogel. In Figure 2C, the wide absorption peaks at 3300 and 3500 cm^−1^ are attributed to the stretching mode of the hydroxyl group (-OH), which can be seen in GO, L-Cys, and GO. The characteristic absorption peaks of chitosan at 2921 and 2870 cm^−1^ relate to C-H asymmetric and symmetric stretching vibrations and at 1651 and 1590 cm^−1^ were due to amine-I and amide-II, respectively [51,52]. The characteristic absorption peaks of maleic anhydrides at 1242 cm^−1^ indicate the stretch of C–O–C, the peak at 1591 cm^−1^ may be assigned as C=C stretching. The absorption peaks at 1857 and 1785 cm^−1^ belong to asymmetrical and symmetrical C=O stretching of cyclic anhydride molecules. In the spectrum of CS/MA hydrogel, a new peak at 1716 cm^−1^ belongs to the amide bond, and the disappeared peak at 1242 cm^−1^ indicates maleic anhydride was successfully grafted to chitosan via the ring opening mechanism. The disappeared peak at 1591 and 2552 cm^−1^ related to the stretching of the C=C group of MA, and the sulfhydryl group of L-Cys indicated L-Cys successfully grated on CS/MA via “thiol-ene” click reaction.

In Figure 2D, the absorption peaks at 1732 cm^−1^ and 1054 cm^−1^ are assigned to the carbonyl stretching of the C=O and C−O groups, which represented the oxidation-containing groups connected to graphene sheets (Appendix A). The absorption peak appears at 1620 cm^−1^ corresponding to the C=C bonds of GO. The band was found at 1067 and 565 cm^−1^ assigned to an asymmetric phosphate group (PO_4_^3−^) bending vibrations, corresponding to β-TCP. Additionally, such a peak was also observed for Z-CS/β-TCP and Z-CS/β-TCP/GO hydrogel. Compared with Z-CS/TCP, the new peak at 1628 cm^−1^ was observed in the spectra of Z-CS/β-TCP/GO, which was attributed to the absorption peak of C=C bonds in aromatic rings, indicating GO was successfully mixed into the zwitterionic hydrogel. The characteristic GO absorption bands of the C=O and C−O groups in the 1732 cm^−1^ and 1054 cm^−1^ regions were overlapping with those of chitosan and MA, which demonstrates GO and TCP were successfully mixed into the Z-hydrogel.

In Figure 2E, the XRD spectra of GO showed a sharp peak at 2θ = 10.8° representing the oxidation of graphite powder (Appendix A), while a broad characteristic peak at 26° relates to its highly organized layer with a decreased interlayer distance and orientation. The diffraction peaks of β-TCP powder matched well the standard JCPDS card No. 0169-009 at 17.08°, 26.24°, 34.45°, 35.92°, 47.41°, 48.47°, 49.80°, 50.89°, and, 53.12° angels, which demonstrate the existence of the calcium carbonate and hydroxyapatite phase.

The peak at 26° of GO and all peaks from β-TCP can be detected in all Z-CS/β-TCP/GO hydrogels. This means β-TCP physically mixed into the Z-CS/β-TCP hydrogel does not change the crystal structures. The peak at 10.8° was decreased and broad with the increase of GO, probably caused by the disappearance of the crystalline structure or the planes to be placed between the CS chains [53].

### 2.5. Cell Proliferation and Viability

The cell proliferation was assessed by CCK-8 staining assay at one, three, and seven days after inoculation. According to Figure 3A, on the initial day of cultured, the OD values of the Z-C group were higher than other groups, and there were no difference changes between the Z-CS/β-TCP/GO-1 and Z-CS/β-TCP/GO-4 groups but were higher than the control and Z-CS/β-TCP group. At three and seven days after inoculation, the cell viability increased accordingly to the GO concentration. The OD value of the Z-CS/β-TCP/GO-2 group was still significantly higher than that of other groups, and all groups were higher than the control group. These results exhibited that the zwitterionic hydrogel has good biocompatibility, are nontoxic and beneficial for rBMSCs proliferation, and L-cysteine increases the ability to promote rBMSCs of maleilated chitosan. Moreover, incorporation of GO into Z-CS will accelerate cell proliferation; however, a high concentration will inhibit the proliferation of rBMSCs, and the 2 mg/mL GO is shown as the optimal concentration for cell proliferation.

The results of live/dead staining (Figure 3B) displayed good biocompatibility with GO incorporation in the zwitterionic hydrogel. All scaffolds enhanced the proliferation of the rBMSCs and displayed no cytotoxicity. Overall, a low dose of GO, particularly GO at a concentration of 2 mg/mL, has a favorable effect on rBMSC proliferation, while a higher concentration of GO (4 mg/mL) may inhibit the proliferation of BMSC [54]. The toxicity of GO depends on the size and concentration [54,55]; small-sized GO has easily been internalized into the cells’ high concentrations and triggers oxidative stress that could cause cell apoptosis [18,56]. The above results indicate that the GO synthesized its owned proper size, and incorporation in the zwitterionic hydrogel decreased the cytotoxicity.

### 2.6. DAPI Staining

To assess the morphological, adhesion, and distribution of rBMSCs, DAPI/TRITC phalloidin staining was utilized after one, three, and seven days of cultivation (Figure 3C). On day one, rBMSCs were dispersed and adhered to the surface of different scaffolds and exhibited a polygon with short filopodia. After three days of culture, the rBMSCs cultured in the Z-CS/β-TCP/GO scaffolds were spread uniformly and well extended compared to the control group. With the concentration of GO increased, the number of rBMSCs increased, had more spindle-like elongation, and a growing protruded pseudopod observed, especially in the Z-CS/β-TCP/GO-2 group. GO promotes rBMSCs to outstretch and exhibit more spreading area [57]. The stretched pseudopodia increased the interactions between cells, which is more important for the spread and proliferation of rBMSCs [58]. After seven days of culture, the rBMSCs revealed long fusiform morphology, intercell overlap growth, and were spreading well. The Z-CS/β-TCP/GO-2 group still showed higher cell proliferation. Zhang et al. [59] showed with increasing GO concentrations, more cell projection area was observed, and the optimal containing of the highest metabolically active was 1 mg/mL. The GO we synthesized had an appropriate size and shape. GO incorporated in zwitterionic hydrogel decreased its release, reduced the cytotoxicity of GO, and promoted cell adhesion, spreading, and proliferation.

The result of DAPI staining was consistent with the CCK-8 and live/dead staining assay. It revealed GO we synthesized, had an appropriate size and shape, and incorporation in zwitterionic hydrogel decreased its release. GO promotes differentiation via controlling cytoskeletal tension [60,61], improving the growth of cellular protrusion [54].

### 2.7. Osteogenic Differentiation of rBMSCs

ALP staining results and alizarin red S staining were used to estimate the early and late osteogenic capacity of rBMSC on different scaffolds. ALP is an essential biomarker of early osteogenic differentiation and participates in the mineralization process. After culturing for seven days, the ALP intensity was stronger than that of the control group (Figure 4A). Additionally, the ALP intensity of the Z-CS/β-TCP/GO-2 group express the strongest compared to the others. After 14 days, the APL intensity increased in all groups, and the Z-CS/β-CTP/GO-2 group was still higher than other groups.

Mineralization is a late marker for osteogenesis [62]. The alizarin red S staining also showed the corresponding results with ALP staining at 14 and 21 days (Figure 4B); more calcium deposition appeared with the increase of GO concentration incorporated in the zwitterionic hydrogel, and the Z-CS/β-CTP/GO-2 group showed the larger-scale mineralized nodules. As in Figure 4C,D, qualitative analysis of the ALP activity and alizarin red S staining showed the same results. This result revealed GO incorporation in zwitterionic hydrogel rapidly accelerated osteogenic differentiation regardless of early or late stage, and the 2 mg/mL GO owned the strongest osteogenic differentiation ability.

### 2.8. Osteoblasts at Molecular Level

For a greater understanding of the effects of zwitterionic hydrogel and GO on osteogenic differentiation, we evaluated the expression of osteogenic genes by qRT-PCR. *RUNX2* is a specific osteoblast transcriptional activator that regulates early osteogenesis differentiation Its target genes included *ALP*, *COL 1*, and *OCN* [63]. *COL 1* is associated with the formation of an extracellular matrix (ECM) at the initial stage of osteogenic differentiation. *OCN* is a late marker of bone formation and correlates with the maturation and mineralization of osteoblasts [64]. β-catenin is a vital marker protein of the Wnt/β-catenin-signaling pathway. The *OPN* is an important noncollagenous protein in bone matrix formation. As shown in Figure 5A–F, the results showed that compared with the control group, the expression levels of *RUNX2*, *ALP*, *COL 1*, *OPN*, and *OCN* were upregulated in all experimental groups (*p* < 0.05) and increased with GO concentration [35], whereas a higher concentration of GO (4 mg/mL) would downregulate the expression of osteogenesis-related genes.

As shown in Figure 5G–J, these results clearly indicated GO improved the expression level of early and late osteo-related protein of rBMSCs by upregulated RUNX2 protein and activating Wnt/β-catenin-signaling, and the effect is dose-dependent regulated [5]. Moreover, western blot results further verified expression of β-catenin, COL-1, and RUNX2 in the Z-CS/β-CTP/GO-2 group was significantly high than in the control group. These results showed GO could promote osteogenic differentiation via the WNT-signaling pathways.

### 2.9. Micro-CT Imagine Analysis

The critical size calvarial defect model with 5 mm in diameter was established to evaluate the ability of bone regeneration in SD rats. Figure 6A shown the Micro-CT scanning of rat calvarial defects at six weeks and twelve weeks after surgery (Appendix A). At six weeks, new bone formation was observed only at the periphery of the defect in the control group, while extensive new bone was found from the border to the center of the defect, and the bone defect became smaller in the Z-CS/β-CTP and Z-CS/β-CTP/GO-2 groups. Twelve weeks later (Figure 6B), more newly formed bone was observed, and the area of the calvarial bone defect was significantly smaller. The new bone formed in the central areas and almost bridged the defect area in the Z-CS/β-CTP and Z-CS/β-CTP/GO-2 groups; the Z-CS/β-CTP/GO-2 group showed a significant new bone and bone trabecula formation. GO incorporated in zwitterionic hydrogel exhibited higher osteogenic potential than others.

As shown in Figure 6C,D, at six weeks, the total BV/TV ratios of the Z-CS/β-CTP and Z-CS/β-CTP/GO-2 groups were 7.03 ± 0.387 and 20.47 ± 0.463, respectively, which were higher than the control group (5.02 ± 0.441). The BMDs of the Z-CS/β-CTP group (0.083 ± 0.004) and the Z-CS/β-CTP/GO-2 group (0.281 ± 0.006) were dramatically higher than the control group. At 12 weeks, BV/TV ratio and BMD all increased, and the Z-CS/β-CTP/GO-2 group was still higher than other groups.

### 2.10. Histological Staining

To histologically investigate new bone regeneration, H&E staining and Masson’s trichrome staining were used for observing new bone regeneration at six and twelve weeks after treatment with different scaffolds (Figure 6E,F). At six weeks, in the control group, a small amount of fibrous tissue regenerated in the peripheral region, whereas fibrous tissue density increased, and some new bone formation was observed in the Z-CS/β-CTP and Z-CS/β-CTP/GO-2 groups. There was more bone matrix in the Z-CS/β-CTP/GO-2 group. At 12 weeks, the fibrous and the thickness of mineralized bone increased in each group. The Z-CS/β-CTP/GO-2 group showed more new bone tissue and new formed bone islands than the control group. The bone tissue was arranged regularly and close to the normal bone structure. These results agree with the in vitro observation that Western blotting and qRT-PCR analyses proved Z-CS/β-CTP/GO-2 scaffolds upregulated the expression of osteogenic protein and gene and promote migration and osteogenesis differentiation of rBMSCs. This indicated zwitterions and GO synergistically promote the proliferation and osteogenic differentiation of rBMSCs in vitro and in vivo.

## 3. Materials and Methods

### 3.1. Materials

The graphene was purchased from Nanjing XFNANO Materials Tech Co., Ltd. (Nanjing, China). Chitosan (CS, degree of deacetylation ≥ 95%,), β-TCP, maleic anhydride, and L-cysteine were obtained from Macklin Biochemical Technology Co., Ltd. (Shanghai, China). The α-minimum essential medium (α-MEM), fetal bovine serum (FBS), penicillin-streptomycin (P/S), and trypsin were purchased from Gibco (ThermoFisher, Shanghai, China). The ascorbic acid, dexamethasone, and β-glycerophosphate were purchased from Sigma Aldrich. All the reagents were used as received without further purification. Malt Sprague–Dawley (SD) rats were purchased from Silaike, Shanghai, China.

### 3.2. Graphene Oxide (GO) Preparation

Graphene oxide (GO) was prepared using the modified Hummers’ method [65] Briefly, 1 g graphite powder was gradually suspended into 230 mL concentrated H_2_SO_4_ (sulfuric acid, 98%) in an ice bath for 30 min, and 6 g KMnO4 was subsequently added to the mixture under constant stirring at 50 °C for 1 h. The mixture was cooled to room temperature, and subsequently, 200 mL of distilled water was added to terminate the reaction, then 5 mL of 30% H_2_O_2_ was added for further oxidation until the color changed from black to bright yellow. GO was obtained by centrifuging the mixture. It was then washed with hydrochloric acid (HCL) and double-distilled water several times until it attained the neutral PH. Finally, GO powder was obtained by freeze-drying the GO suspension.

### 3.3. Fabrication of CS/MA Hydrogel

The schematic representation of the synthetic scheme for zwitterionic hydrogel is shown in Figure 7. MA was dissolved into 4 mL dimethyl sulfoxide (DMSO) solution, and then, chitosan (molar ratio between maleic anhydride and chitosan at 1:1) was added into the solution followed by constant stirring for 1 h.

At the same time, three distinct concentrations (1%, 2%, and 4% mg/mL) of GO were dispersed in DMSO via ultrasonication for 30 min until completely uniformly dispersion. Subsequently, β-TCP nanoparticles, based on a weight ratio of 3:7 to CS/MA, were dispersed in GO solution, and ultrasonic treatment at room temperature was continued to evenly load β-TCP on GO sheets. Afterward, it was gradually added to the obtained CS/MA solution drop by drop, and the entire system was ultrasonically treated for another 2 h, then transferred to a 24-well plate and dried in an oven for 24 h at 37 °C. The hydrogel was washed with a large amount of distilled water to remove the unreacted MA and DMSO from the hydrogel network.

### 3.4. Fabrication of Z-CS/β-CTP/GO Hydrogel

To integrate zwitterionic material, the obtained hydrogel was immersed in L-Cys solution protected by nitrogen gas for 24 h (Appendix A), then taken out and dialyzed against pure water. Finally, the hydrogel was frozen at −80 °C for 24 h and then lyophilized and freeze-dried for another 24 h to obtain the Z-CS/β-TCP/GO scaffolds. The scaffolds containing various concentrations of GO (0, 1%, 2%, 4% mg/mL) were denoted as Z-CS/β-TCP, Z-CS/β-TCP/GO-1, Z-CS/β-TCP/GO-2, and Z-CS/β-TCP/GO-4, respectively.

### 3.5. Material Characterization

#### 3.5.1. Scanning Electron Microscopy (SEM) Analysis

The morphology of the Z-CS/β-TCP/GO scaffolds was observed via the scanning electron microscope (SEM) (ZEISS Gemini 300, Oberkochen, Germany); these samples were mounted on an aluminum stub and coated with gold by a sputter coater. The images were captured at a voltage of 15 kV.

#### 3.5.2. FT-IR and X-ray Diffraction (XRD)

Fourier transform infrared spectra (FT-IR) analysis was used to examine the chemical structure of these samples. The freeze-dried hydrogels were ground into powder and pressed into disks by mixing with potassium bromide (KBr) powder. The spectra were measured with an FT-IR spectrophotometer (Thermo Scientific Nicolet iS20, Waltham, MA, USA) over the spectral range of 4000–400 cm^−1^.

To further prove the existence of the GO and β-TCP in the zwitterionic hydrogel, the characteristic phase and crystallinity were investigated via x-ray diffraction spectroscopy (Bruker D8 Advance) using 40 mA and 40 kV current. Data were recorded in the scanning range from 10° to 80° (2θ degrees) at a step size of 1°/min.

#### 3.5.3. Porosity Measurement

The porosity of the fabricated scaffolds was measured by the liquid displacement method. Each scaffold was divided into equal weights, and the known volume of ethanol in the graduated cylinder was considered as *V*_1_. After the scaffolds were immersed in the ethanol and total saturation, the total volume of ethanol and the scaffold were recorded as *V*_2_. Then, the scaffold was removed, and the residual volume of ethanol was defined as *V*_3_. The porosity ratio of the scaffold was calculated according to the formula as follows.
$$Porosity\left(\%\right)=\frac{\left(V_{1}-V_{3}\right)}{\left(V_{2}-V_{3}\right)}\times 100\%$$

#### 3.5.4. Swelling Capacity

The swelling ratio of the scaffolds was estimated using a conventional gravimetric procedure. Each dry scaffold was weighed (*W*_0_) and immersed in PBS at 37 °C. At definite time points of 0.5, 1, 3, 6, 12, and 24 h, the samples were taken out and placed on filter paper to wipe extra water. Then, the swollen weight of the sample was measured (*W*_1_). The swelling ratio was determined with the equation as follows:$$Swelling\;ratio\left(\%\right)=\frac{\left(W_{1}-W_{0}\right)}{W_{0}}\times 100\%$$

#### 3.5.5. Degradation Studies

The initial weight of each lyophilized scaffold was recorded as W_0_, then immersed in the PBS containing lysozyme (500 U/mL) and incubated at 37 °C for 3, 7, 14, 21, and 28 days. After setting the time point, scaffolds were taken out, washed with distilled water, and freeze-dried. The remaining weight was noted as *W_d_*. The degradation of the sample was calculated from the following formula: $$Degradation\;ratio\left(\%\right)=\frac{\left(W_{d}-W_{0}\right)}{W_{0}}\times 100\%$$

#### 3.5.6. Mechanical Properties

The mechanical properties of the Z-CS/β-TCP/GO hydrogel scaffolds were analyzed using an electric universal testing machine (Instron, Norwood, MA, USA). The scaffolds were cut into cylinder samples of 10 mm in diameter and 15 mm in height and tested at a speed of 1 mm/min until 80% maximal deformation of the construct was reached. The compressive modulus was calculated from the linear region of the stress–strain curve. Young’s modulus was measured via the 0.2% strain offset linear slope method based on compressive stress at 50% strain.

### 3.6. Cell Proliferation and Cell Viability

The scaffolds were disinfected by soaking in 75% ethanol for 30 min and rinsed in sterilized PBS three times. Rat bone marrow mesenchymal stem cells (rBMSCs) were isolated and cultured from bone marrow of four-week-old Sprague–Dawley (SD) rats. Passage 4 of the rBMSCs was seeded on the surface of the sterilized scaffolds at a density of 5 × 10^4^ /mL in 24-well plates. At given time points (1, 3, 7 days), the medium was eliminated, and 10% CCK-8 was added to each well. After being incubated in 5% CO_2_ at 37 °C for 2 h, the samples were measured at 450 nm by a microplate reader. Cell viability was assessed using live/dead staining. After being transferred to a 2 μM Calcein-AM and 4 μM PI solution, the cells were cultured in a CO_2_ cell incubator for 30min. Confocal microscopy (Nikon, Tokyo, Japan) was used to visualize live cells (green) and dead cells (red).

### 3.7. ALP Staining and ALP Activity Assay

Early osteoblast differentiation was evaluated by ALP staining and ALP activity assay. rBMSCs were seeded at 8 × 10^3^ cells/well in a 24-well. After 24 h, the solution was replaced with an osteogenic differentiation medium (supplied with 10 mM/L β-glycerophosphate, 0.1 mM/L dexamethasones, and 0.05 mM/L ascorbic acid) to initiate differentiation. Cells cultured in a medium without any scaffolds were served as the control group. At seven and fourteen days, the cells were stained via BCIP/NBT Alkaline Phosphatase Kit (Beyotime, Shanghai, China) according to the manufacturer’s protocol. The ALP activity was measured via ALP activity assay kits (Beyotime, Shanghai, China), and recorded by a microplate reader at 520 nm.

### 3.8. Alizarin Red Staining

The mineralization of rBMCs cocultured with different scaffolds was detected by alizarin red staining S (Beyotime, Shanghai, China) at 14 and 21 days. Briefly, at each time point, cells were washed with PBS and fixed in 4% PFA for 20 min, then attained with Alizarin Red S solution. Microscopic images were required via an inverted microscope (Nikon, Japan). To quantify, the absorbance was measured at 562 nm by a microplate with 10% cetylpyridinium chloride.

### 3.9. Cell Morphology

For studying the spreading and morphology of rBMCs at one, three, and seven days, the cultured cells with different scaffolds were washed three times with PBS and fixed in 4% paraformaldehyde for 10 min and permeabilized with 0.1% TtitonX-100 solution in PBS for 5 min. TRITC-Phalloidin and 4′,6-diamidino-2-phenylindole (DAPI) diluted in 1% BSA was adopted to stain cytoskeleton (red) and nuclei (blue), and then, the staining results were captured using a confocal fluorescence microscope (SP8; Leica, Heidelberg, Germany).

### 3.10. Quantitative Real-Time PCR Analysis

Total RNA was extracted from rBMSCs by Trizol reagent (Beyotime, China) and was reversely transcribed into complementary DNA (cDNA) using Prime Script RT Master Mix (Takara, Shiga, Japan). Quantitative real-time PCR analysis was performed using PowerUp SYBR Green Master Mix (Thermofisher, USA) to evaluate the expression of osteogenic differentiation-related genes of *RUNX2*, *ALP*, *COL 1*, *OPN*, *β-Catenin*, and *OCN* with *GAPDH* used as an internal reference. Data were analyzed according to the comparative Ct (2^−ΔΔCt^) method, and the fold change was calculated as target genes normalization with the control. The primer sequences are shown in Appendix A.

### 3.11. Western Blot Analysis

To investigate the protein levels, the protein was extracted by RIPA lysis buffer and quantified by BCA Protein Assay Kit (Beyotime, China). The same amount of protein was separated on SDS-PAGE gel by electrophoresis and transferred to polyvinylidene difluoride (PVDF) membranes. After being blocked in 5% skim milk for 1 h at room temperature, the membrane was incubated with primary antibodies of RUNX-2, COL 1, β-Catenin, and GAPDH at 4 °C overnight and incubated using secondary antibodies for 2 h. The band signals were visualized via an electrochemiluminescence detection agents (ECL) reagent and quantified by Image J V1.8.0 software. The expression level of each target protein was normalized against GAPDH.

### 3.12. In Vivo Study

Considering the above results, such as mechanical properties and in vitro cell proliferation and osteogenic differentiation, we chose Z-CS/β-CTP/GO-2 hydrogel as the optimum scaffold for vivo study.

All animal experiments were performed in accordance with the guidelines of the Animal Care and Experimental Ethical Committee of Tongji University.

#### 3.12.1. Micro-CT Imagine Analysis

After six and twelve weeks post-surgery, rat calvaries were harvested and scanned using a micro-CT system (Burker, Salbruken, Germany) to assess bone repair in the defect area. Then, the 3D images were reconstructed by Mimics 16.0 software (Materialise, Belgium), and the percentages of new bone volume/tissue volume (BV/TV) and bone mineral density (BMD) in the calvarial defects were quantitatively analyzed.

#### 3.12.2. Histological Analysis

The new bone regeneration rate at the defect site was assayed by immunohistochemical staining. After micro-CT imaging, rat calvaries were decalcified in 15% (*w*/*v*) ethylene diamine tetraacetic acid (EDTA), dehydrated in a graded series of alcohol, embedded in paraffin, and sequentially sectioned at 5 μm thick on a coronal plane using a paraffin microtome (Leica EG 10060). Histological analyses were subjected by using hematoxylin and eosin (H&E) and Masson’s trichrome stains to observe new tissues. The images were photographed by an optical microscope (Nikon, Japan).

### 3.13. Statistical analysis

Statistical analyses were performed using GraphPad Prism 8.0 (GraphPad Software Inc., San Diego, CA, USA), and all experiments were carried out with at least three replicates. Results were expressed as mean ± standard deviation (SD). All data were analyzed by one-way analysis of variance (ANOVA), and *p* < 0.05 were considered statistically significant.

## 4. Conclusions

In summary, we have successfully synthesized novel zwitterionic hydrogel incorporation of GO and crosslinked with L-cysteine via click chemical. Specifically, this novel zwitterionic hydrogel exhibited good porosity, swelling, degradation ratio, and osteogenic differentiation ability. Adding GO in zwitterionic hydrogel synergistic improved physicochemical properties and induced the attachment, proliferation, and osteogenic differentiation of rBMSCs. The optimum formulation of GO was 2 mg/mL. In vivo study, micro-CT scanning, and histology staining confirmed GO incorporation increased the promotion of new bone formation in critical size of rat calvarial defects. Overall, the novel GO-incorporated zwitterionic hydrogel exhibited extensive employment in bone tissue engineering.

## Figures and Tables

**Figure 1 ijms-24-02691-f001:**
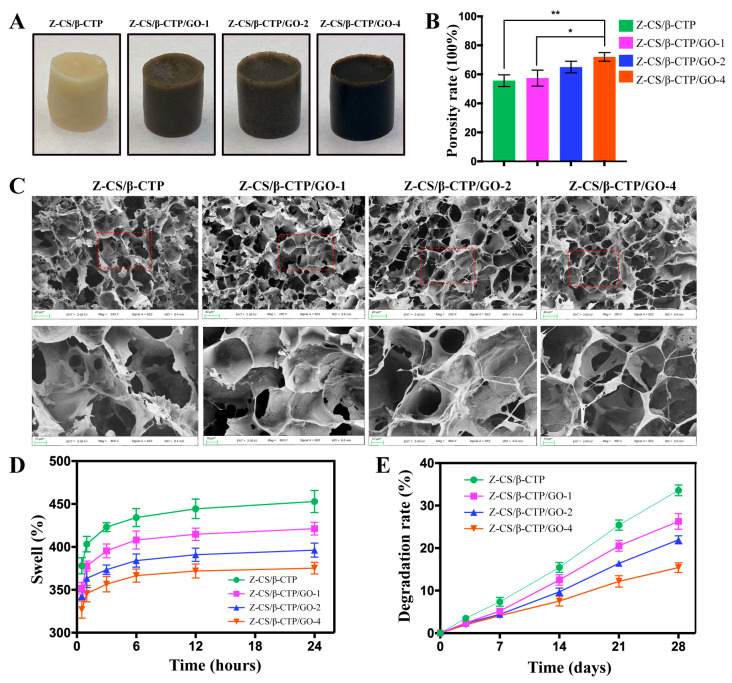
(**A**) Optical images of hydrogels. The color changed gradually from milky white to black with the increase of GO concentration. (**B**) Pore sizes of different scaffolds increased with the increase of GO concentration (* *p* < 0.05, ** *p* < 0.01). (**C**) SEM images of scaffolds with different concentrations of GO. The red dotted boxes in the upper panels were enlarged in the lower panels. (**D**) Swelling rate and (**E**) degradation rate decreased with increasing GO concentration.

**Figure 2 ijms-24-02691-f002:**
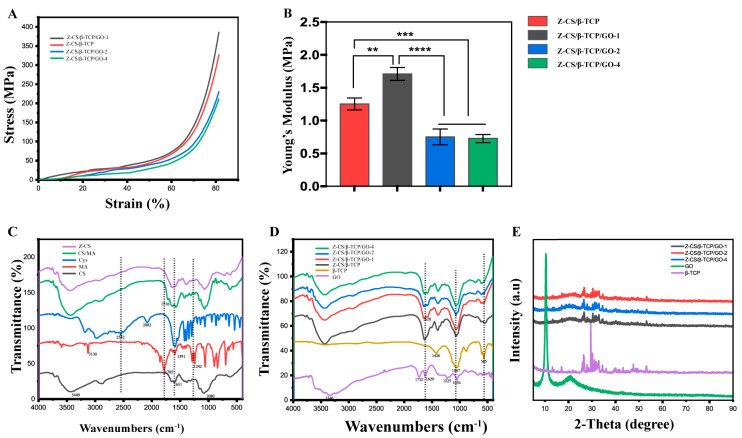
Mechanical properties of the zwitterionic hydrogel. (**A**) Stress—strain curve. The concentration of GO at 1mg/mL showed the strongest stress—strain curves. (**B**) Young’s modulus of the zwitterionic scaffolds at different concentrations of GO ** *p* < 0.01, *** *p* < 0.001, and **** *p* < 0.0001). FT-IR spectra of (**C**) CS, MA, Cys, CS/MA, Z-CS, and (**D**) GO, β-TCP, Z-CS/β-TCP, Z-CS/β-TCP/GO-1, Z-CS/β-TCP/GO-2, Z-CS/β-TCP/GO-4. XRD patterns of (**E**) GO, β-TCP, Z-CS/β-TCP/GO-1, Z-CS/β-TCP/GO-2, Z-CS/β-TCP/GO-4 of different samples.

**Figure 3 ijms-24-02691-f003:**
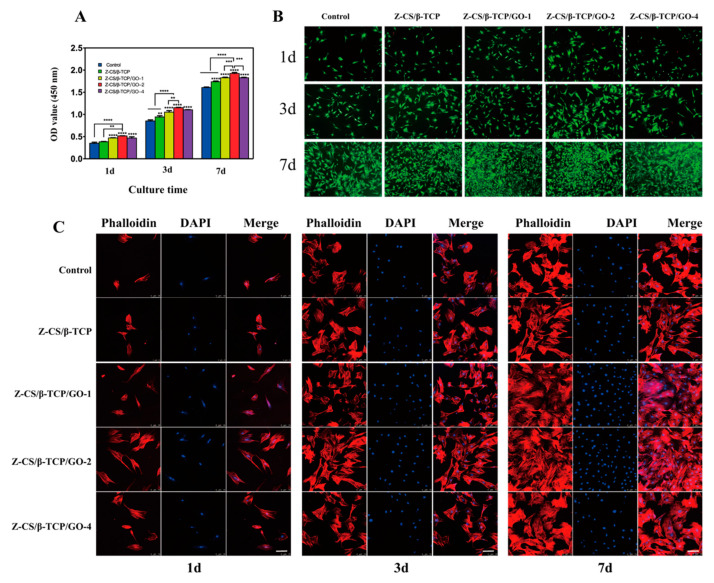
Proliferation, viability, and morphology of cells cultured with different scaffolds for one, three, and seven days. (**A**) CCK-8 assay (** *p* < 0.01, *** *p* < 0.001, and **** *p* < 0.0001). (**B**) Live/dead staining: green (live), red (dead), Scale bars, 100 μm. (**C**) Fluorescent images of rBMSCS morphologies: blue (nucleus), red (cytoskeleton), Scale bars, 300 μm.

**Figure 4 ijms-24-02691-f004:**
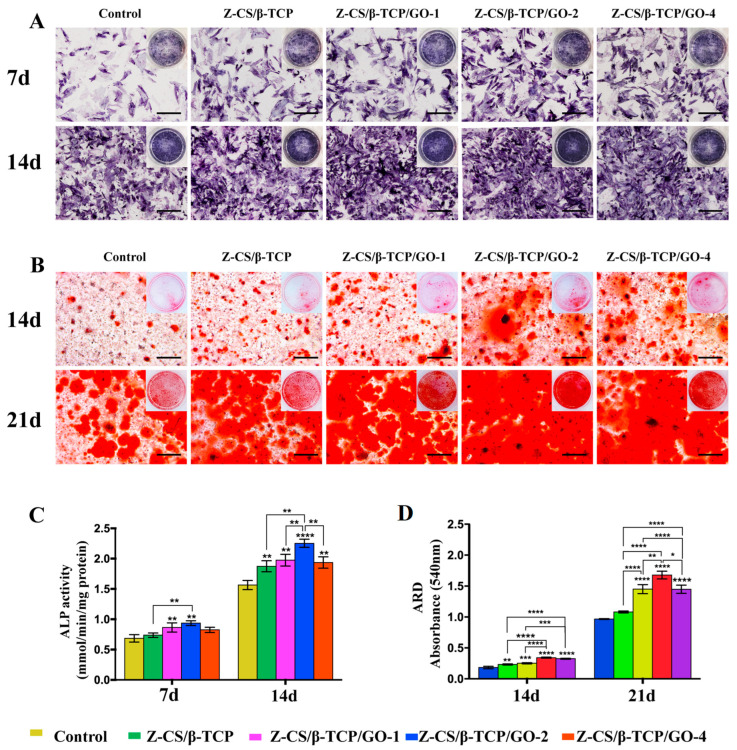
rBMSc osteogenic activity of rBMSc with different scaffolds. (**A**) ALP straining staining of rBMSCs cultured with different scaffolds for seven and fourteen days, scale bars, 200 μm. (**B**) Alizarin red S stained cultured with different scaffolds for 14 and 21 days, scale bars, 200 μm. (**C**) ALP activity. (**D**) Calcium nodules (* *p* < 0.05, ** *p* < 0.01, *** *p* < 0.001, and **** *p* < 0.0001).

**Figure 5 ijms-24-02691-f005:**
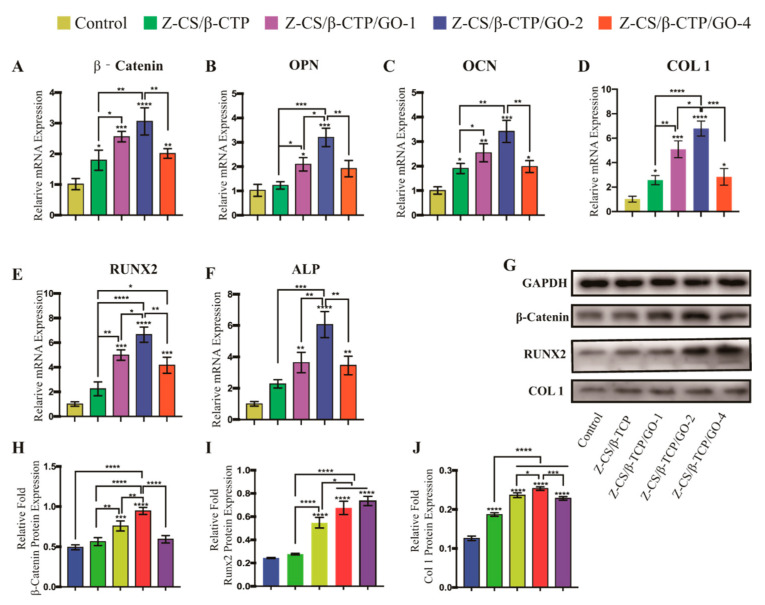
Expression levels of osteogenic genes β-catenin (**A**), *OPN* (**B**), *OCN* (**C**), *COL 1* (**D**), *RUNX2* (**E**), and *ALP* (**F**) tested by qRT-PCR. Osteogenic proteins (**G**) analyzed by western blotting. (**H**–**J**) Quantification results of β-catenin, RUNX2, and COL-1 (* *p* < 0.05, ** *p* < 0.01, *** *p* < 0.001, and **** *p* < 0.0001).

**Figure 6 ijms-24-02691-f006:**
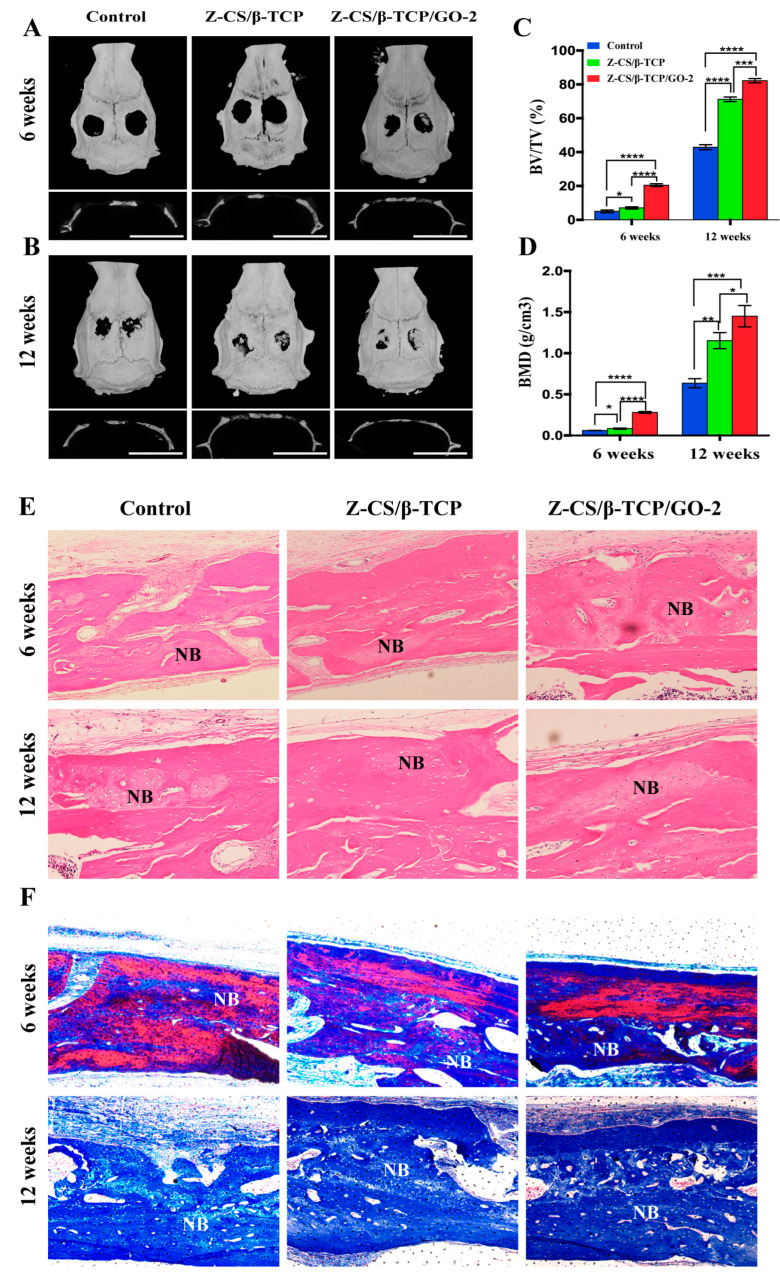
Osteogenic differentiation in vivo. Bone repairing model by Micro-CT at (**A**) six weeks (scale bars, 10 mm) and (**B**) twelve weeks post implantation (scale bars, 20 mm). (**C**) Quantitative comparison of bone mineral density (BMD) and (**D**) bone volume to total volume (BV/TV). Histological analysis of bone regeneration in different zwitterionic hydrogel at six and twelve weeks (* *p* < 0.05, ** *p* < 0.01, *** *p* < 0.001, and **** *p* < 0.0001). (**E**) H&E staining and (**F**) Masson’s trichrome staining showed fibrous tissues were regenerated in the control group. The newly formed bone with fibrous tissues was obvious in the Z-CS/β-CTP group. In the Z-CS/β-CTP/GO-2 group, a mount of new bone was observed. (NB: newly formed bone. Magnification ×200.)

**Figure 7 ijms-24-02691-f007:**
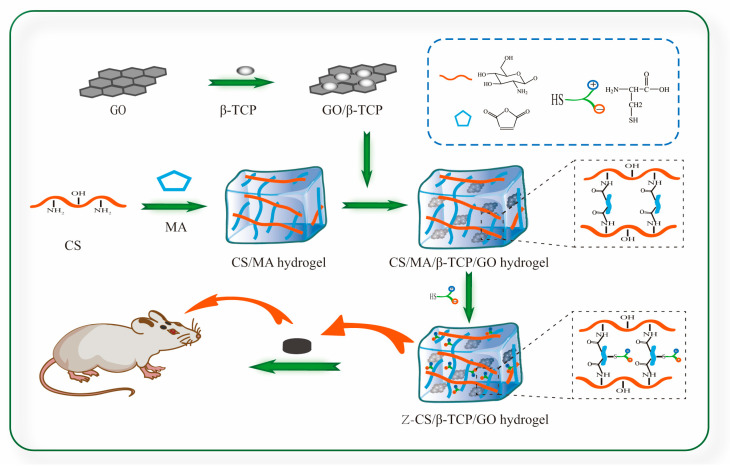
Scheme of the synthetic zwitterionic hydrogel.

## Data Availability

The data that support the findings of this study are available from the corresponding author upon reasonable request.

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
