# Peer review of "A Novel Zwitterionic Hydrogel Incorporated with Graphene Oxide for Bone Tissue Engineering: Synthesis, Characterization, and Promotion of Osteogenic Differentiation of Bone Mesenchymal Stem Cells"

_ijms, 2023, doi:10.3390/ijms24032691_

Round 1
Reviewer 1 Report
The authors have prepared a zwitterionic hydrogel incorporated with graphene oxide for bone tissue engineering. However, the novelty of the work is poor, and the advantages of zwitterionic hydrogel as bone tissue engineering repair were not well described. Reviewer also think the paper is very rough and not suitable for publication now.
1. The figures in the whole paper are in the wrong order.
2. Captions in Figures are incomplete, such as Figure 3 and Figure 7.
3. The Figure 2 is missing.
4. In line 31, there is a syntax error. The comma can’t join two separate sentences.
5. In Fig. 2C, the scales of SEM images are too fuzzy and I can’t know what each image represents.
6. In line 93-101, “no obvious change in the pore structure can be seen” and “the addition of GO improves surface roughness and pore interconnectivity” contradict each other. And in line 108-109, you say “The porosity of scaffolds increased with the GO concentration increased”.
7. In line 134-135, “SEM image” should change to “SEM images” and it doesn't say what does * stand for.
8. In line 124, “Fig. 4E” should change to “Fig. 2E”.
9. In line 138-139, “the stress-strain curves were enhanced greatly” can’t from the Fig. 3A.
10. In line 148, “Fig. 3B” should change to “Fig. 3C”.
11. In line 187, “FT-IR(B-C) and XRD (D)” should change to “FT-IR(C-D) and XRD (E)”.
12. When the concentration of GO was 2 mg/ml, the mechanical properties decreased significantly. Why is 2 mg/ml the best concentration? And cited the literature “Zhang et al.[60] showed with increasing GO concentrations, more cell projection area was observed and the optimal containing of the highest metabolically active was 1 mg/ml.”
Author Response
Dear Reviewers,
Thank you very much for taking the time to review this manuscript. I really appreciate all your comments and suggestions! Please find my itemized responses below and my revisions/corrections in the re-submitted files.
The authors have prepared a zwitterionic hydrogel incorporated with graphene oxide for bone tissue engineering. However, the novelty of the work is poor, and the advantages of zwitterionic hydrogel as bone tissue engineering repair were not well described. Reviewer also think the paper is very rough and not suitable for publication now.
Response:The introduction of the original manuscript of our paper does not make clear the new ideas and importance. In view of this, we have strengthened the abstract and introduction to highlight innovation. The innovation of this article lies in that we first applied maleilated chitosan and zwitterion to bone tissue engineering. We use maleic anhydride (MA) as a cross-linking agent to modified chitosan (CS) to improve the mechanical strength and decrease the degradation rate of CS. Previous covalent cross-linking has the disadvantages of poor strength and fast degradation. Maleic anhydride as a crosslinking agent is no cytotoxicity, and avoided using toxic substances as crosslinking agents. The C=C double bond of maleic anhydride provides a site for L-cysteine grafted on via click chemistry. When GO was incorporated in this Zwitterionic scaffold, the zwitterionic scaffold slows down the release rate and reduces the cytotoxicity of GO. Zwitterions and GO synergistically promote the proliferation and osteogenic differentiation of rBMSCs. In the follow-up study, we can graft different substances on maleic anhydride to endow zwitterionic scaffold with different characteristics.
- The figures in the whole paper are in the wrong order.
Response:It seems that it is an error in the submission system and has been changed in the revised version
- Captions in Figures are incomplete, such as Figure 3 and Figure 7.
Response:We are very sorry for our incomplete writing and it is rectified.
- The Figure 2 is missing.
Response:We have checked and revised the Figure 2
- In line 31, there is a syntax error. The comma can’t join two separate sentences.
Response:We have checked and revised it
- In Fig. 2C, the scales of SEM images are too fuzzy and I can’t know what each image represents.
Response:We are very sorry,we has changed the SEM images in the revised version and added detailed description
- In line 93-101, “no obvious change in the pore structure can be seen” and “the addition of GO improves surface roughness and pore interconnectivity” contradict each other. And in line 108-109, you say “The porosity of scaffolds increased with the GO concentration increased”.
Response:In line 93-101, “no obvious change in the pore structure can be seen”, We are very sorry for our negligence of the explanation and we have deleted this sentence.
- In line 134-135, “SEM image” should change to “SEM images” and it doesn't say what does * stand for.
Response: We have change SEM image” to “SEM images” and added description to what does * stand for.
- In line 124, “Fig. 4E” should change to “Fig. 2E”.
Response:We are very sorry for our incorrect writing and it is rectified
- In line 138-139, “the stress-strain curves were enhanced greatly” can’t from the Fig. 3A.
Response:Our statement is not accurate enough and it has been revised to delete the “greatly”.
- In line 148, “Fig. 3B” should change to “Fig. 3C”.
Response:We are very sorry for our incorrect writing and it is rectified
- In line 187, “FT-IR(B-C) and XRD (D)” should change to “FT-IR(C-D) and XRD (E)”.
Response:We are very sorry for our incorrect writing and we have rectified the Captions in Figures
- When the concentration of GO was 2 mg/ml, the mechanical properties decreased significantly. Why is 2 mg/ml the best concentration? And cited the literature “Zhang et al.[60] showed with increasing GO concentrations, more cell projection area was observed and the optimal containing of the highest metabolically active was 1 mg/ml.”
Response:Although when the concentration of GO is 2 mg/ml, the mechanical properties decreased, it still maintains high mechanical strength and is still suitable for bone tissue engineering. Zhang et al. proved that the optimal concentration was 1mg/ml. However, in our study,2 mg/ml of GO had the strongest proliferation and osteogenic differentiation ability in this hydrogel scaffold. Probably due to the GO we synthesized having appropriate size and shape,and decreased the release and reduced the cytotoxicity of GO when incorporated in this Zwitterionic hydrogel. When considering the mechanical properties and proliferation ability, we finally chose the optimal concentration is 2mg/ml of GO. We added the description in line 252.
With best regards,
Qigong Wang

Reviewer 2 Report
This paper is about a novel zwitterionic hydrogel using MA research and has a normal structure.
In the introduction, each basic material of the final product that the researcher wants to develop is well explained, but the purpose of their fusion, structural explanation related to it, and research purpose are not sufficient. The research purpose of the author who can seamlessly connect the description of each material is required.
B-MSC stem cells were used, but there is no description of harvesting the cells. You must accurately indicate which animal line was selected and where the cells were obtained from.
The purpose of using MSCs rather than using single cells related to existing bone is probably to investigate cell differentiation and proliferation in detail. Please connect animal test results based on western blot results. The author's description of animal testing (H&E staining and Masson's trichrome staining) is insufficient.
Author Response
Dear Reviewers,
Thank you very much for taking the time to review this manuscript. I really appreciate all your comments and suggestions! Please find my itemized responses below and my revisions/corrections in the re-submitted files.
- In the introduction, each basic material of the final product that the researcher wants to develop is well explained, but the purpose of their fusion, structural explanation related to it, and research purpose are not sufficient. The research purpose of the author who can seamlessly connect the description of each material is required.
Response:In this study, we use maleic anhydride (MA) as a cross-linking agent to modified chitosan (CS) to improve CS mechanical strength and decrease its degradation rate, and L-Cys as a zwitterionic, can graft on maleilated chitosan (MSC) via click chemistry.
β-tricalcium phosphate (β-TCP) as an inorganic, was added into the scaffold material to simulate the organic and inorganic components of the human body, and improve the mechanical strength of the scaffold material.
Graphene oxide (GO) can promote cell proliferation and osteoblastic differentiation, but high concentrations will induce cytotoxicity. In this new zwitterionic hydrogel, we added zwitterions to reduce the cytotoxicity of GO, and zwitterions and GO synergistic promoting cell proliferation and osteogenic differentiation.
- B-MSC stem cells were used, but there is no description of harvesting the cells. You must accurately indicate which animal line was selected and where the cells were obtained from.
Response:Rat bone marrow mesenchymal stem cells (rBMSCs) were isolated and cultured from bone marrow of 4-week-old Sprague-Dawley (SD) rats. We have added it in line 470-471 and detailed descripted the isolation and cultivation methods in the supporting information.
- The purpose of using MSCs rather than using single cells related to existing bone is probably to investigate cell differentiation and proliferation in detail. Please connect animal test results based on western blot results. The author's description of animal testing (H&E staining and Masson's trichrome staining) is insufficient.
Response:We have redescribed the results of H&E staining and Masson's trichrome staining, and connect animal test results with qRT-PCR and western blot results shows that zwitterions and GO synergistic promoting cell proliferation and osteogenic differentiation in vivo and in vitro.
With best regards,
Qigong Wang
